
# A protocol for quantifying mono- and polysaccharides in seawater and related saline matrices by electro-dialysis (ED) – combined to HPAEC-PAD

**Sebastian Zeppenfeld[1], Manuela van Pinxteren[1], Anja Engel[2], Hartmut Herrmann[1,*]**

1 Atmospheric Chemistry Department (ACD), Leibniz-Institute for Tropospheric Research (TROPOS), Leipzig, Germany

2 GEOMAR Helmholtz Centre for Ocean Research Kiel, Kiel, Germany

*Correspondence to*: Hartmut Herrmann (herrmann@tropos.de)

## Abstract

An optimized method is presented to determine free (DFCHO) and combined monosaccharides (CCHO) in saline matrices, such as oceanic seawater, Arctic ice core samples or brine using a combination between desalination with electro-dialysis and high performance anion exchange chromatography coupled to pulsed amperometric detection (HPAEC-PAD). Free neutral sugars, such as glucose and galactose, were found with 95-98% recovery rates. Free amino sugars and uronic acids were strongly depleted during electro-dialysis at pH=8, but an adjustment of the pH could result in higher recoveries (58-59% for amino sugars at pH=11; 45-49% for uronic acids at pH=1.5). The applicability of this method for the analysis of CCHO was evaluated with standard solution and real seawater samples compared with another established desalination method using membrane dialysis. DFCHO in real field samples from different regions on earth ranged between 11-118 nM and CCHO between 260-1410 nM. This novel method potentially contributes to a better understanding of biogeochemical processes in the oceans and sea-air transfer processes of organic matter into the atmosphere during further research studies.

## Introduction

The majority of organic matter (OM) in oceanic seawater can be assigned to the chemical classes of proteinogenic amino acids, carbohydrates and lipids (Benner and Kaiser, 2003; Kaiser and Benner, 2009; Kuznetsova and Lee, 2002; Marty et al., 1979; Skoog et al., 1999; Wakeham et al., 1997). Previous studies uncovered that combined amino acids are the most abundant organic substances in fresh particles sinking



within the water column (Wakeham et al., 1997), although more recent studies indicate that
carbohydrates may be equally abundant (Cisternas-Novoa et al., 2019), while hydrolysable carbohydrates
dominate the chemical composition of dissolved organic matter (DOM) (Kaiser and Benner, 2009). Marine
carbohydrates also appear in high concentrations in other related saline matrices, such as ice cores, brine
and melt ponds in the Arctic (Ewert and Deming, 2013; Underwood et al., 2013; Zeppenfeld et al., 2019).
Hence, a reliable analysis of carbohydrates is essential for understanding biogeochemical processes in the
(Arctic) ocean and their impact on Earth's atmosphere.
Most marine carbohydrates exist as polysaccharides or combined sugars (CCHO), which are linear or
branched chains of monosaccharides, including deoxy sugars, amino sugars and uronic acids. In living
marine microorganisms including prokaryotes, polysaccharides assume their functions as structural
compounds or as energy storage (Skoog and Benner, 1997). Storage carbohydrates mainly consist of
glucose, such as laminarans and other glucans, while structural heteropolysaccharides (e.g. galactans) such
as occurring in algal cell walls can contain a lot of galactose, mannose and rhamnose (McCarthy et al.,
1996). Furthermore, an elevated release of polysaccharides by phytoplankton, mostly of gelatinous nature,
has been associated to stress situations, such as a deficiency of nutrients, freezing or fluctuating water
potential (Berman-Frank et al., 2007; Bianchi and Canuel, 2011; Borchard and Engel, 2012, 2015; Ittekkot
et al., 1981; Krembs et al., 2002; Krembs and Deming, 2008). These exuded polysaccharides are relatively
depleted in glucose and galactose and mainly contain acidic sugars, fucose, rhamnose and arabinose in
their chemical structure (Borchard and Engel, 2012; Passow, 2002). Even though polysaccharides are
ubiquitous in nature, a latest study revealed that the individual sugar pattern is different between algae
and terrestrial plants (Hepp et al., 2016) and may allow a source apportionment of carbohydrates in
seawater.

Dissolved free monosaccharides (DFCHO) have been found to form another fraction of marine
carbohydrates (Engel and Händel, 2011; Ittekkot et al., 1981; Kirchman et al., 2001). DFCHO are considered
to be either directly released by phytoplankton cells or be the product of enzymatic degradation of
polysaccharides (Pakulski and Benner, 1994). In most studies, DFCHO are found in lower concentrations
than CCHO, since marine microbes utilize them with high turnover rates (Sakugawa and Handa, 1985;
Thornton et al., 2016). From the concentrations of DFCHO, or rather the ratio between CCHO and DFCHO,
information about in situ activities of local phytoplankton and bacteria in the seawater can be obtained
(Pakulski and Benner, 1994; Sakugawa and Handa, 1985). Recently, correlations between the



concentrations of free glucose in Arctic surface water samples and their ice nucleating activity (INA)
suggested a potential link between the formation of INA and marine carbohydrates (Zeppenfeld et al.,

64   2019).

At the ocean surface, wind and wave interactions lead to bubble bursting. The emitted sea spray aerosol
contains marine carbohydrates, including hydrogels, which contribute to the chemical and physical
properties of these particles (Bigg and Leck, 2008; Frossard et al., 2014; Hawkins and Russell, 2010;
Rosenørn et al., 2006). They have been detected in particles at different maritime regions on earth,
including the North Atlantic, the Arctic and Antarctica (Barbaro et al., 2015; Frossard et al., 2014; Gao et
al., 2011, 2012; Leck et al., 2013; Russell et al., 2010). However, understanding the quantitative fluxes of
marine carbohydrates from the ocean to the atmosphere is still challenging, since chemical analysis of
sugars in seawater strongly suffers from matrix effects, especially caused by sea salt.
The concentrations of individual monosaccharides in seawater, related saline matrices and aerosol
particles can be determined with different kind of chromatographic methods, such as high performance
liquid chromatography and gas chromatograph. These methods require a quite difficult sample
preparation, including a labor-intensive derivatization step (Panagiotopoulos and Sempéré, 2005). In the
last decades, high performance anion exchange chromatography coupled to pulsed amperometric
detection (HPAEC-PAD) has been established as a reliable alternative, since it facilitates a sensitive
quantification of sugar compounds both in seawater and in airborne particles without a prior derivatization
(Iinuma et al., 2009; Panagiotopoulos and Sempéré, 2005; van Pinxteren et al., 2012; Skoog and Benner,
1997). However, the presence of sea salt in seawater samples strongly affects the chromatographic
performance of the HPAEC-PAD and needs to be removed before analysis.
Several procedures are available for the desalination of seawater. The desalination using anion exchange
resins AG2-X8 and the cation exchange resin AG50W-X8 exhibits strong drawbacks such as the complete
loss of charged sugars (amino sugars, uronic acids) and quite low recovery rate of neutral sugars between
20-80% depending on the individual monosaccharide (Borch and Kirchman, 1997; Mopper et al., 1992;
Rich et al., 1996). The use of silver cartridges (Dionex OnGuard II Ag/H Cartridges) is faster and easier, but
requires very expensive consumables and the capacity of removable sea salt per cartridge is strongly
limited (Mopper et al., 1992; Panagiotopoulos and Sempéré, 2005). The desalination applying dialysis
membranes achieves reproducible and very high recovery rates of hydrolysable polysaccharides (> 90%).
However, this method does not allow the analysis of DFCHO, since these small molecules pass the
membrane during dialysis (Engel and Händel, 2011).




Electro-dialysis is a fast way to remove ions by applying an electrical field. The use of two different chemo-
selective ion exchange membranes allows the exclusive removal of small anions, or cations respectively.
Hence, uncharged small substances (neutral DFCHO) and macromolecules (CCHO) can, in principle, be
recovered in high quantities. Electro-dialysis is being used for the desalination of salty water to generate
potable water, the denitrification of wastewater and soil remediation (Gain et al., 2002; Ottosen et al.,
2000; Sadrzadeh and Mohammadi, 2008; Tsiakis and Papageorgiou, 2005; Wisniewski et al., 2001). For
analytical sample preparation, electro-dialysis has been reported as a powerful desalination, e.g. for the
analysis of DOM and marine neutral DFCHO (Josefsson, 1970; Koprivnjak et al., 2009; Mopper et al., 1980;
Vetter et al., 2007; Wirth et al., 2019). However, biases, which have hitherto not been discussed in this
analytical context, can occur during the application of electro-dialysis due to the osmotic and electro-
osmotic loss of water, the migration and diffusion of monosaccharides and the appearance of sudden pH
changes.

Within the present study, a novel protocol for the desalination of seawater samples and related saline
samples, applying electro-dialysis and HPAEC-PAD is presented, accounting for the described biases. This
method with a low need of consumables allows the analysis of individual monosaccharides with (CCHO)
and without hydrolysis (DFCHO). This developed technique was applied to analyze a diverse set of
carbohydrates in different kinds of ambient seawater samples.

## 113   2. Experimental

2.1 Chemicals and materials
Prior to the analysis of carbohydrates in seawater, all used laboratory glassware had been washed with
ultrapure water (conductivity >18.2 MΩ·cm) thoroughly and pre-heated in a muffle furnace at 550°C for 4
h. All plastic equipment was washed in 10% HCl solution and washed with ultrapure water three times.
For calibrating the HPAEC-PAD and determining the recovery of individual monosaccharides, a mixed stock
solution was prepared from fucose (Roth, 95%), galactosamine (Sigma, 99%), rhamnose (Sigma, 99%),
arabinose (Sigma, 99%), glucosamine (Fluka), galactose (Fluka, 99%), glucose (99%), xylose (Fluka, 99%),
mannose (Fluka,99%), fructose (Aldrich, 99%), ribose (Aldrich, 98%), muramic acid (Sigma, 95%),
galacturonic acid (Sigma-Aldrich, 97%), glucuronic acid (Sigma, 97%), mannuronic acid (Sigma, 90%).



Synthetic seawater samples were made of commercially available sea salts (Sigma). The salinity in practical
salinity units (PSU) and the pH of water aliquots was measured by using a conductivity meter (pH/Cond
3320, WTW).


2.2 Field samples
Seven different real seawater samples, one ice core from Arctic sea ice and two brines collected within
Arctic ice cores (**Table 1**) were measured and used for evaluating recovery rates and comparability of the
method presented here. These saline samples were collected during different campaigns of our
department and kept stored at -20°C. All sampling bottles had been rinsed with dilute hydrochloric acid
(10% v/v) prior to the campaign. Field blanks (ultra-pure water filled up in sampling bottles during the
campaign) were collected during each campaign and treated in the same way as the samples.

**Table 1** Sampling details of discussed saline samples including seawater samples (SWS), ice cores (IC) and
brine (B). SML stands for surface microlayer *(Engel et al., 2017)*. *(Wendisch et al., 2019)*

| Saline sample (SS) | Location | Sampling date | Campaign | Latitude | Longitude | Depth (m) |
|---|---|---|---|---|---|---|
| SWS 1 | Tropical Atlantic | 13.11.2011 | Cape Verde | 16.935°N | 024.915°W | 0 (SML) |
| SWS 2 | Tropical Atlantic | 13.11.2011 | Cape Verde | 16.935°N | 024.915°W | 2 |
| SWS 3 | Raunefjorden | 16.05.2011 | Raunefjorden | 60.274°N | 005.181°E | 2 |
| SWS 4 | North Atlantic | 07.05.2012 | ANT-XXVIII-5 | 33.3°N | 013.5°W | 0 (SML) |
| SWS 5 | Arctic Ocean | 13.07.2017 | PS 106* | 81.229°N | 018.744°E | 1 |
| SWS 6 | Arctic Ocean | 14.07.2017 | PS 106* | 81.015°N | 026.883°E | 1 |
| SWS 7 | North Sea | 25.05.2017 | PS 106* | 57.288°N | 005.213°E | 1 |
| IC 1 | Arctic Ocean | 12.06.2017 | PS 106* | 81.824°N | 011.571°E | 0-0.8 |
| B 1 | Arctic Ocean | 12.06.2017 | PS 106* | 81.824°N | 011.571°E | 0-0.8 |
| B 2 | Arctic Ocean | 12.06.2017 | PS 106* | 81.824°N | 011.571°E | 0-1.5 |


2.3 The electrodialysis system

The centerpiece of the PCCell Micro Bench Electrodialysis system for small sample volumes consisted of
three separated compartments (**Figure 1**): The sample compartment was an open chamber that was filled
up with 9 ml of the standard solution or seawater sample. The functionalized anion exchange membrane
(quaternary ammonium aliphatic polyether) and cation exchange membrane (sulfonated aromatic



polyether) bordered this compartment on both sides. Depending on their chemical properties, the
membranes allowed exclusively the migration of either positively or negatively charged ions. The contact
surface with the sample was 7.8 cm$^2$ for each membrane. For maintaining the conductivity within the
system and receiving the sea salt from the sample, the next compartment contained the concentration
circuit, a 16 g·L$^{-1}$ NaCl solution (Merck). This solution was circulated at a rate of 60 ml·mL$^{-1}$.Two end
membranes on each side divided the concentration circuit from the third department including the
electrodes. The mixed metal oxide (MMO) anode was made of a titanium base body coated by $RuO_2$, $IrO_2$
and $TiO_2$. The MMO cathode was based on stainless steel. The electrodes were permanently surrounded
by a circulating 0.25 M $Na_2SO_4$ (Fluka) electrolyte circuit for avoiding unwanted redox reactions, e.g. the
generation of corrosive elemental chlorine from chloride. Spacers were inserted between each membrane
for keeping the electrolyte and concentration circuits well mixed. The sample solution was homogenized
with a pipette during each desalination. The electrolyte and the concentration solutions were regularly
renewed. The maximal electrical current $I_{max}$ within the ED cell was adjusted by an automatic online

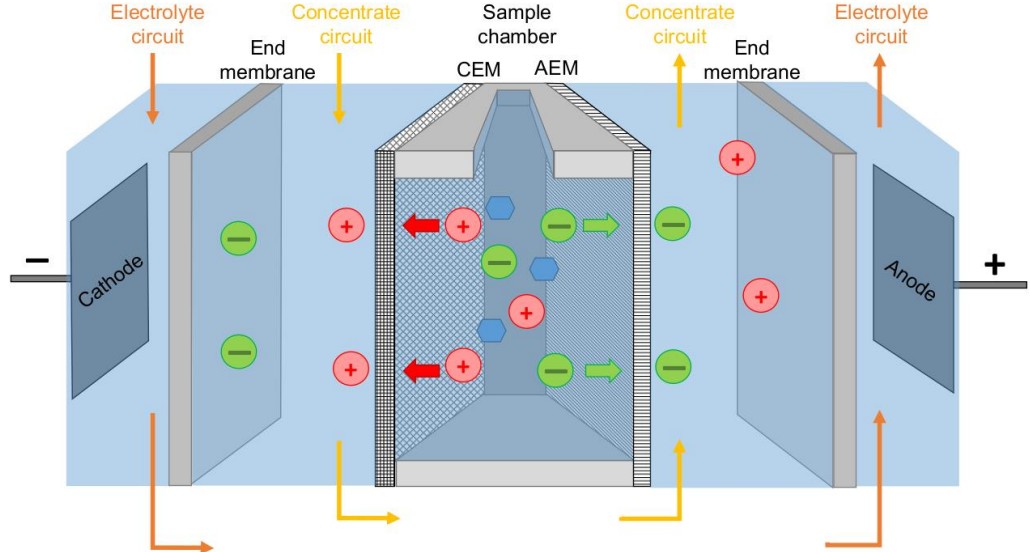

**Figure 1** *Schematic setup of the used ED cell. Red circles represent cations, green circles anions, and blue hexagons are carbohydrates. CEM and AEM stand for cation exchange membrane and anion exchange membrane respectively.*

adaption of the voltage, which never exceeded 25 V. The desalination was stopped when the electric
current dropped to a value of 0.20 A.
The used ion-exchange membranes have a quite long lifetime as long as they are not damaged
mechanically. However, very high attention needs to be given to remove residues of previous desalinated





samples in order to avoid carry-over phenomena and obtain good reproducibility. Hence, every time
before a new sample was desalinated, the sample chamber was always first exposed to ultrapure water
for ten minutes and then flushed once with an aliquot of the new sample, which was disposed after.

2.4 HPAEC-PAD system
HPAEC-PAD was applied for the analysis of marine carbohydrates in seawater samples. Here, we used an
Dionex ICS-3000 ion chromatography system coupled to an autosampler AS-1 as it has been already
described for the analysis of saccharidic biomass burning markers in atmospheric particles (Iinuma et al.,
2009). Several neutral monosaccharides, amino sugars and uronic acids were separated on a Dionex
CarboPac 20 analytical column (3x150mm) combined with a Dionex CarboPac PA20 guard column
(3x30mm), which was permanently temperature conditioned at 30°C. The separation of these saccharides
was conducted by applying the gradient profile shown in **Table 2**, which was an adaption to the elution by
(Meyer et al., 2008). Neutral and amino sugars eluted within the first 19 min at 4 mM NaOH. By adding
sodium acetate, sugar acids eluted and organic and inorganic contaminants were flushed from the column.
After the removal of remaining acetate by 250 mM NaOH, the system was equilibrated at 4 mM NaOH for
the next sample injection. The flow rate of the eluent was 0.5 mL·min$^{-1}$. The retention times, peak widths
and resolution factors of the measured monosaccharides are shown in **Table 3**. For the injection of a
sample aliquot, a 25 µL loop was used. Each sample was measured as a duplicate and each standard as a
triplicate. Limits of detection (LOD) of individual monosaccharides were ranging between 2-12 nM, which
is in good agreement with literature (Engel and Händel, 2011; Panagiotopoulos and Sempéré, 2005).

For the preparation of eluents A-D, filtered ultra-pure water (conductivity >18.2 MΩ·cm) was degassed
with helium for 20 min. Eluents A and B were made by adding a defined volume of low-carbonate NaOH
solution (Fisher Chemical, 50% w/w) to the degassed water. Eluent C was prepared by dissolving sodium
acetate (Thermo scientific, anhydrous) in ultra-pure water, filtering it through a nylon membrane (0.2 µm,
Thermo Scientific), degassing the solution with helium for 20 min and adding the corresponding volume
of NaOH solution.




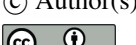



**Table 2** *Gradient profile applied on the CarboPac PA20 column during HPAEC-PAD analysis*

| Time (min) | %A (250 mM NaOH) | %B (20 mM NaOH) | %C (1 M Na-acetate/ 250 mM NaOH) | %D (H$_2$O) |
|---|---|---|---|---|
| 0 | 0 | 20 | 0 | 80 |
| 19 | 0 | 20 | 0 | 80 |
| 20 | 5 | 0 | 15 | 80 |
| 35 | 5 | 0 | 15 | 80 |
| 38 | 20 | 0 | 40 | 40 |
| 39 | 100 | 0 | 0 | 0 |
| 44 | 100 | 0 | 0 | 0 |
| 45 | 0 | 20 | 0 | 80 |
| 78 | 0 | 20 | 0 | 80 |


**Table 3** *Averaged peak characteristics of monosaccharide standards separated by the presented elution averaged over 24 runs.*
*Resolution factors are calculated between peak and following peak in chromatogram.*

| | Retention time (min) | Peak width (min) | Resolution factor |
|---|---|---|---|
| Fucose | 3.79±0.01 | 0.27 | 10.5 |
| Galactosamine | 7.28±0.01 | 0.40 | 1.0 |
| Rhamnose | 7.70±0.02 | 0.42 | 1.1 |
| Arabinose | 8.15±0.02 | 0.42 | 2.2 |
| Glucosamine | 9.23±0.02 | 0.56 | 2.1 |
| Galactose | 10.33±0.02 | 0.51 | 2.8 |
| Glucose | 11.81±0.03 | 0.56 | 3.4 |
| Xylose | 13.89±0.03 | 0.65 | 0.8 |
| Mannose | 14.43±0.04 | 0.77 | 2.8 |
| Fructose | 16.52±0.05 | 0.75 | 2.0 |
| Ribose | 18.20±0.05 | 0.93 | 13.2 |
| Muramic acid | 25.84±0.01 | 0.23 | 9.0 |
| Galacturonic acid | 29.27±0.02 | 0.53 | 4.6 |
| Glucuronic acid | 31.49±0.02 | 0.45 | 2.8 |
| Mannuronic acid | 32.83±0.02 | 0.49 | |



2.5 Protocol for the analysis of DFCHO and CCHO in seawater and other saline samples
Stored frozen samples were defrosted in a fridge at 4°C. 9 ml of the filtered sample (0.2 µm, Millex, PTFE)
was desalinated with electrodialysis as described above. In the end of each desalination, electro-(osmotic)
water loss was replenished with ultra-pure water and mixed thoroughly. In order to analyze free amino
sugars or uronic acids, pH could be adapted with concentrated HCl or 1 M NaOH.
A concentration step using a vacuum concentrator (MiVac) at 55°C allowed the detection of low
concentrated DFCHO, as it occurs in most seawater samples. For this purpose, a round-bottom glass vial
was filled with an aliquot of 6 ml desalted sample, which was weighted empty and filled. After reaching a
remaining volume of less than approximately 600 µl, the glass vial was weighted again (in order to calculate
the concentration factor) and the concentrated aliquot was pipetted in the autosampler vial for HPAEC-
PAD analysis. This step allowed a decrease of LOD by a factor of 10. Each sample was prepared and
measured as duplicate.
In order to measure CCHO, marine polysaccharides need to be cleaved into their monomeric compounds
by acid hydrolysis. We applied the optimized conditions described by Engel and Händel (2011) with slide
modifications. An aliquot of 1 ml desalted sample was hydrolyzed with hydrochloric acid (HCl
concentration in sample= 0.8 M) in pre-heated (550°C, 4 h) glass ampules for 20 h at 100°C. Neutralization
was performed by evaporating all liquid under vacuum at 55°C until dryness. The dry residue was dissolved
in 700 µl ultra-pure water, treated with a vortex homogenizer (IKA MS 3 basic) and filled in the
autosampler vial for HPAEC-PAD analysis. Each sample was prepared and measured as duplicate.

2.6 Parameter optimization and assessment of method
Impact of osmosis and electro-osmosis during ED desalination
For quantifying the loss of water in the sample due to osmosis and electro-osmosis, a synthetic sea salt
solution was pipetted into the desalination chamber, which was desalinated for 0, 5, 10, 15, 20 and 25 min
with a voltage of 25 V and a maximal current of 0.6 A. After the lapse of time, the total remaining volume
was pipetted quantitatively into a glass vial and weighted (Mettler Toledo, XS105 DualRange). These
measurements were repeated for four different sea salt solution (10, 20, 30 and 40 PSU) and as triplicate
for each time. The recovery of the sample mass was calculated as the ratio between the mass after the
corresponding desalination time and the averaged mass after 0 min.



Recovery of DFCHO within the ED membrane system
Standard addition experiments with real seawater were performed, for quantifying the recovery of
monosaccharides due to diffusion and migration under consideration of all matrix effects. For that reason,
sample 7 was filtered (0.2 μm, Millex, PTFE) and spiked with a sugar standard mix (neutral sugars, amino
sugars, uronic acids) resulting in a concentration increase of 10 μg L$^{-1}$ and 100 μg L$^{-1}$.These samples were
desalinated using electrodialysis ($I_{max}$=0.6 A, stop at 0.2 A). In the end of each run, (electro-) osmotic water
loss was either replenished or not, and the sample directly measured with the HPAEC-PAD. These
measurements were repeated as triplicates for each concentration. In order to account for possible
wasting phenomena, repetitions were performed with new membranes, as well with membranes, which
already had been used for some time before. Given recovery rates for neutral monosaccharides are the
average of the results for 10 μg L$^{-1}$and 100 μg L$^{-1}$. For sugar acids and amino sugars, only the averaged
recovery rates for 100 μg L$^{-1}$ are given for avoiding determinations close to the LOD.
In order to investigate the influence of pH on the migration of charged monosaccharides, this experiment
was repeated for three different pH values: At pH=8 (natural pH of seawater), pH of 1.5 (acidified with
concentrated HCl) and pH of 11 (addition of 1 M NaOH). Since high pH in seawater leads to precipitation
of hydroxides of alkaline earth metals, an additional filtering (0.2 μm) was performed for these runs.

Recovery of CCHO within the ED membrane system
Recovery experiments were performed with solutions and a suspension of the polysaccharide standards
sodium alginate (Aldrich), laminarin from Laminaria digitate (Sigma) and cellulose powder from spruce
(Fluka) at natural pH. Stock solutions were added to filtered sample SWS 7 resulting in concentrations of
10 mg L$^{-1}$. Aliquots of 1 ml with and without desalinations were hydrolyzed (HCl 0.8 M, 100°C, 20 h) and
neutralized by evaporation with the vacuum concentrator (55°C) until dryness. The residue was
reconstituted in 700 μL, treated with a vortex homogenizer (IKA MS 3 basic) and filled in the autosampler
vial for HPAEC-PAD analysis. Recovery rates were calculated as a ratio between the determined
monosaccharide concentrations after hydrolysis of the standard solutions with and without desalination.
In order to compare our method on the recovery of CCHO with another established method, aliquots of
four seawater samples were treated following the electro-dialysis protocol presented here and the
protocol by Engel and Händel (2011) using membrane desalination, an acid hydrolysis with HCl (0.8 M,

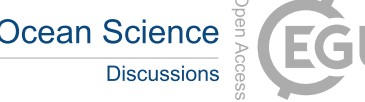

100°C, 20h), neutralization by evaporation (nitrogen, 50°C) and an elution on a Dionex CarboPac PA10
column.

## 3. RESULTS AND DISCUSSION

A reproducible quantification of carbohydrates in seawater samples using HPAEC-PAD requires a prior
removal of disturbing sea salt. Here, we present electrodialysis as a reliable desalination method, its
parameter optimization and the discussion of arising phenomena resulting in a protocol for the analysis of
marine carbohydrates.

### 3.1 Kinetics and efficiency of desalination

During the desalination of seawater by electrodialysis, anions and cations migrate through an electrical
field and pass chemo-selective membranes. Depending on their electrical charge, they move either to the
positively charged anode or to the negatively charged cathode. In this process, the salt flux through the
membranes $j_S$ (mol·m$^{-2}$·s$^{-1}$), which determines the desalination time, is proportional to the applied
electrical current $I$ (Han et al., 2017; Vanoppen et al., 2015). **Figure 2** shows the current within the used
ED system and the salinity of the seawater sample during a typical desalination of an artificial seawater
sample (40 PSU) for two different applied maximal currents $I_{max}$ within the system. For almost the entire
desalination run, the current $I$ was maintained at $I_{max}$ due to automatic adjustment of the voltage. During
this time, the salt flux was approximately constant. Towards the end of the desalination, when almost all
salt ions were removed, the current dropped down and the salt flux became lower. Since a direct salinity
measurement was not possible in the sample chamber without contaminating the sample, the end of each
desalination was defined, when the current $I$ reached a value of 0.2 A. At this point, the salinity of the
sample typically ranged between 0.2 and 0.4 PSU, which was found to be sufficiently low for the
carbohydrate analysis at the HPAEC-PAD. This reduction in salinity represents an overall desalination of
more than 99% of the initial salt concentration. A desalination reaching a salinity below 0.1 PSU was
possible, but was not necessary for this application and would have resulted in longer desalination times.
Consequently, for minimizing the required desalination time, a high $I_{max}$ is favorable.
However, it was observed that the application of an $I_{max}$ of more than 0.8 A during the desalination,
resulted in a strong rise of the pH and a white precipitation in the (synthetic) seawater solution, apparently



due to the formation of hydroxides of alkaline earth metals. This uncontrolled precipitation strongly
disturbed the efficiency of the desalination and the reproducibility of the carbohydrate measurements and
caused a scaling of the membranes. Previous studies explained these unfavorable changes of pH by a
strong concentration polarization at the membranes surface leading to water splitting to $H^+$ and $OH^-$ ions,
when a certain limiting current is exceeded. This phenomenon has been preferably observed at anion
exchange membranes with quaternary amino groups in the presence of divalent cations, such as $Mg^{2+}$ and
$Ca^{2+}$ (Cowan, 1962; Martí-Calatayud et al., 2018; Ottosen et al., 2000). The described phenomena
exclusively occurred when (synthetic) seawater was desalinated and not during the desalination of NaCl
standard solutions. This finding shows the importance of performing parameter optimization tests with

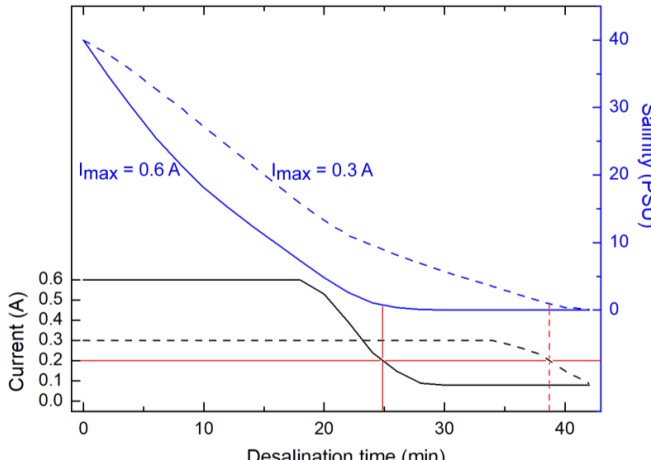

**Figure 2** *Measured current within the ED system and salinity of a synthetic seawater sample (40 PSU) versus desalination time with two different maximal applied currents $I_{max}$ (0.6 A solid line, 0.3 A dashed line). The red line represents the current and the corresponding salinity, when desalination was stopped.*

synthetic seawater standards that include all important seawater constituents such as divalent cations. In
summary, the optimum maximal current $I_{max}$ of 0.6 A was found for the used ED system for avoiding scaling
effects and performing desalination as fast as possible.

3.2 Possible biases during the application of electrodialysis
During the application of ion exchange membranes, the passive transport of water (*osmosis*) and solutes
with a low molecular weight (*diffusion*), such as dissolved monosaccharides, can occur triggered by a
concentration gradient between the sample and concentration channels (Galama et al., 2014; Galier et al.,
2012). By operating an electrical field, the active transport of charged molecules (*migration*) and hydrated



water (*electro-osmosis*) takes place (Galama et al., 2014; Galier et al., 2012). While osmosis and electro-
osmosis induce an unavoidable loss of water and hence of the total volume of the sample, diffusion and
migration of the analytes result in a loss of analyzable molecules. All these phenomena might falsify the
determined concentration of the analytes in the sample and need to be characterized for an accurate
sample preparation for the analysis of marine carbohydrates.

Osmotic and electro-osmotic transport of water
Osmosis describes the passive transport of free water molecules through a partially permeable membrane
caused by large differences of the osmotic pressures between the concentrate circuit and the sample
solution (Sata, 2007). The direction and the quantity of the water transport depends on the residence time
$t_R$ of the sample solution within the membrane system, the difference between the concentrations of
solutes in the sample solution and the concentration circuit $(c_s - c_c)$ and membrane specific parameters,
such as the osmotic water transfer coefficient the membrane area and the membrane thickness (Galama
et al., 2014). The quantitative effect of osmosis can be reduced by minimizing $t_R$ and $(c_s - c_c)$. Hence, $c_c$ was
set at 16 g NaCl $L^{-1}$, which is approximately in the middle between the concentrations of a typical seawater
sample before (30-39 PSU) and after the desalination (0.2-0.4 PSU) for balancing the positive and negative
contribution of osmosis on the total sample volume during a typical desalination. Under these conditions,
a maximum loss of 3% sample volume was observed in the described ED system due to osmosis.
In aqueous solutions, water molecules form a hydration shell around ions (Ohtaki and Radnai, 1993).
Whenever ions pass through membranes during electrodialysis, a cotransport of these hydrating water
molecules occurs, known as electro-osmosis (Galama et al., 2014). The electroosmotic water transfer $j_W$
($m^3 \cdot m^{-2} \cdot s^{-1}$) is proportional to the salt flux $j_S$ in the system and can be expressed by formula (Eq-I) with the
molar volume of water $V_M$ ($1.8 \cdot 10^{-5} \cdot m^3 \cdot mol^{-1}$) and the salt hydration number $n_H$ (mol water$\cdot mol^{-1}$ salt)
(Galier and Balmann, 2015; Han et al., 2015).
$$j_W = n_H \cdot V_M \cdot j_S \qquad\qquad\qquad\qquad\qquad\qquad\qquad\qquad \text{(Eq-1)}$$
The salt hydration number of NaCl, as the major compound of sea salt, has been reported with values
between 11 and 14 (Han et al., 2015; Rutgers and Hendrikx, 1962; Singlande et al., 2006; Walker et al.,





2014). Assuming a NaCl concentration of 30 g·L$^{-1}$ and $n_H$ to be 14, a maximal reduction of the sample
volume onto 87 % due to electro-osmosis is expected, additionally to osmosis.
The recovery of the sample volume due to electro-osmosis and osmosis during the desalination was
characterized for four different salinities for the used ED system (**Figure 3**). During the active removal of
sea salt, electro-osmosis is the dominating force causing the water loss in the sample. The electroosmotic
water loss is continual as long as the salt flux stays constant. However, in the final stages of each
desalination, the salt flux decreases and consequently the electroosmotic water transfer decreases, too.

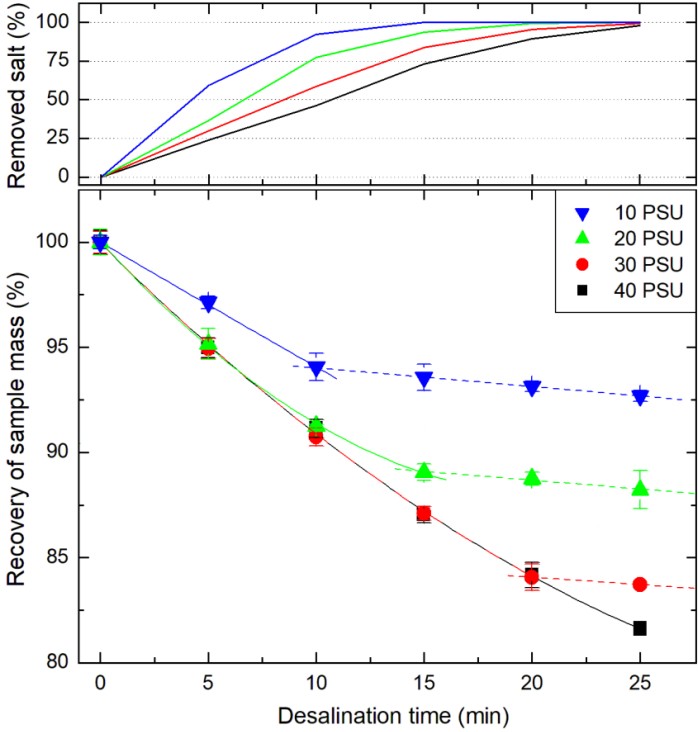

*Figure 3* *Combined effect of electro-osmosis and osmosis (solid lines) and osmosis (dashed lines) on the recovery of sample mass as a function of the desalination time within the described membrane system ($I_{max}$=0.6 A, $c_c$=16 g NaCl L$^{-1}$) for artificial sea salt solutions with four different initial salinities.*

For a synthetic seawater sample with a salinity of 30 PSU, 84% of sample mass was recovered. This is in
good agreement with the estimation mentioned above considering the additional contemporaneous
contribution of osmosis of about 2-3%. Once the sea salt is removed, osmotic water transfer remains at
constant rate of approximately 0.1%·min$^{-1}$.
The overall water loss resulting from osmosis and electro-osmosis needs to be taken into account since it
falsifies the determined concentrations of marine carbohydrates. For its compensation, the chamber was



replenished with ultra-pure water in the end of each desalination until the initial sample volume was
reached. This procedure was performed with a maximal overall error of 0.5%.

Analysis and recovery of DFCHO in seawater samples
The recovery of neutral monosaccharides during electro-dialysis is impacted by diffusion and convection
processes (Galier and Balmann, 2015). Additionally, free amino sugars and uronic acids migrate through
an electrical field due to their charge and pass the ion exchange membranes. Recovery tests were
performed with standard solutions spiked to a real seawater sampled, which have been typically reported
for seawater samples (Kirchman et al., 2001; Mopper et al., 1980; Skoog et al., 1999; Zeppenfeld et al.,
2019). Recovery rates of neutral sugars (Glc, Man, Xyl, Gal, Ara, Fuc, Rha, Fru) ranged between 95-98% at
the natural pH of seawater (approx. pH=8) (**Table 4**). Hence, the overall impact of diffusion and convection
on the recovery of monosaccharides is quite low for the short contact time with these membranes.
However, a higher loss of neutral monosaccharides due to diffusion was observed, when the sample
solution remained within the membrane system for a longer period of time, which calls for a fast
desalination. An overestimation of the determined concentrations was avoided by performing a correction
of the water loss in the end of each desalination. Charged monosaccharides were found with much lower
recoveries of 25-31% for uronic acids and 16-19% for amino sugars at pH=8. This is due to their weak
acidic/basic properties ($pK_a$ (amino sugars) = 7.6-8.5 (Bichsel and von Gunten, 2000; Sinnott, 2007), $pK_a$
(uronic acids) = 3.3-3.5 (Kohn and Kovác, 1978)) and hence their partially ionic state, which makes them
migrate through the electrical field. However, a low pH can protonate the carboxylic group of uronic acids
and a high pH deprotonates the amino group of amino sugars for reducing this effect. Here, we found that
an initial pH of 1.5 before desalination could increase the recovery of free uronic acids up to 45-49%, while
a high pH of 11 resulted into a higher recovery of free amino sugars up to 58-59%. The recovery of neutral
sugars seemed to be quite unaffected within the range of the tested pH, with the exception of fructose,
which was recovered with 89% at pH 1.5, certainly due to its instability within acid conditions. To our
knowledge, here we report for the first time a method, which allows a possible determination of free
amino sugars and uronic acids in saline matrices, such as seawater or the brine from Arctic sea ice.








**Table 4** *Recovery of individual free monosaccharides (neutral, amino sugars, uronic acids) after desalination with electro-dialysis*
*including correction of (electro-) osmotic water loss. n.d.= not determined*

| Monosaccharide | Recovery rate (%) | | |
| --- | --- | --- | --- |
| | $pH_{Start} = 1.5$ | $pH_{Start}= 8$ (seawater) | $pH_{Start}= 11$ |
| Galactose | 97.6±1.0 | 96.0±0.9 | 97.2±1.4 |
| Fucose | 97.2±2.4 | 97.9±1.6 | 97.0±1.0 |
| Glucose | 96.9±1.5 | 97.2±1.4 | 97.1±2.1 |
| Mannose | 95.8±1.8 | 95.8±1.3 | 95.8±2.1 |
| Xylose | 95.6±2.3 | 95.9±2.2 | 95.2±2.0 |
| Rhamnose | 93.5±1.6 | 94.2±1.6 | 95.1±1.4 |
| Arabinose | 93.4±2.3 | 95.1±1.1 | 94.5±1.9 |
| Fructose | 89.3±3.4 | 94.6±2.7 | 94.1 ±2.2 |
| Glucuronic acid | 48.7±2.7 | 31±1.9 | n.d. |
| Mannuronic acid | 44.9±0.6 | 25±1.9 | n.d. |
| Muramic acid | 24.7±1.7 | n.d. | n.d. |
| Glucosamine | 1.5±0.4 | 18.9±1.2 | 59±3.2 |
| Galactosamine | 1.2±0.3 | 15.9±1.5 | 58±3.4 |


Analysis and recovery of CCHO in seawater samples with standard polysaccharides
Recovery experiments with standard solutions of common polysaccharides were performed with and
without desalination by electrodialysis. The neutral, water-soluble polysaccharide laminarin was
recovered with 91.0±5.4%. The acidic polysaccharide alginic acid could be recovered with 93.2±5.3%. Even
though alginic acid might move within the electrical field due to its acidic molecular structure, its molecular
weight does not allow passing the membrane and does not leave the sample solution. Standard
desalination experiments with a suspension of the water insoluble cellulose, which could represent the
fraction of particulate polysaccharides, resulted in much worse recoveries of 48±19%. The reason for this
high, less reproducible loss of polysaccharide was likely caused by sedimentation within the sample
chamber. Engel and Händel (2011) described adsorption processes during the desalination with dialysis
membranes and tackled this problem with sonification of the membranes. However, sonification could not
be applied in our gadget. In this study, flushing the chamber several times with a defined volume of ultra-
pure water after desalination and reuniting the washing water with the desalinated sample could increase
the yield of cellulose up to 85.2±6.9% under consideration of dilution factors. This procedure was not



found to be feasible, since a dilution of a natural sample reduced the sensitivity of low concentrated sugars
in seawater in the analysis by HPAEC-PAD. Rather, we recommend electro-dialysis only for the application
at filtered samples (dissolved compounds), while particulate organic matter might be better analyzed from
filters after filtration.

Comparison of electro-dialysis and dialysis method for the determination of CCHO in seawater samples
In order to evaluate the presented procedure for the analysis of CCHO, comparison studies have been
performed measuring four ambient seawater samples (SWS 1-4) with the established membrane dialysis
protocol after Engel and Händel (2011) and with the here presented method. **Figure 4** shows the results
of the individual monosaccharides after hydrolysis with HCl. Major concentrated sugars, such as glucose,
galactose and xylose/mannose, were determined at similar concentrations. Furthermore, a good
agreement was observed for minor concentrated sugars, such as fucose and galactosamine. Mild
discrepancies were found for rhamnose and arabinose, which appeared in higher concentrations after the
electro-dialysis method, and glucosamine, which was determined at lower concentrations. These
variations might be explained by statistical uncertainties or co-elution of unknown substances. In
summary, the here presented method using electro-dialysis has shown to be in good agreement with the
established membrane dialysis method regarding the analysis of CCHO. In addition, the electro-dialysis

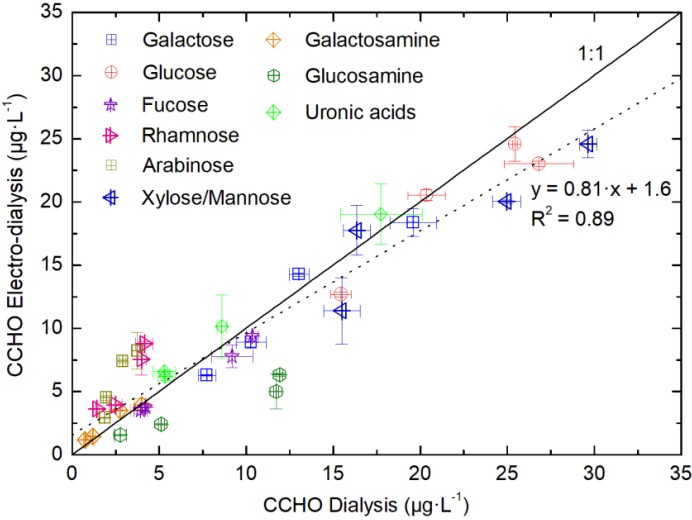

**Figure 4** Determined monosaccharide concentrations in four seawater samples (SWS 1-4) after hydrolysis (CCHO) comparing the desalination by dialysis (Engel&Händel, 2011) and electro-dialysis (presented in this study).

offers the major advantage of analysing the full spectrum of DFCHO as well - which comprise a group of
hardly investigated but potentially important marine compounds.

3.3 Chromatographic performance with HPAEC PAD after desalination.
Several kinds of saline samples were desalinated with electro-dialysis and analyzed on the CarboPac PA20
column. **Figure 5** shows some examples for DFCHO and CCHO chromatograms in a brine and a seawater
sample after desalination with electro-dialysis. The insufficient chromatographic separation of mannose
and xylose using previous kinds of analytical columns has been frequently described in literature (Borch
and Kirchman, 1997; Engbrodt, 2001; Engel and Händel, 2011; Kirchman et al., 2001). Therefore, xylose
and mannose have frequently given only as sum concentrations. The elution of the sugars on a CarboPac20
column, applied in the present study, strongly improved the separation between the both sugars mannose
and xylose (resolution factor=0.8), and allowed the individual determination of these two sugars.
However, most of the analyzed samples showed high concentrations of xylose, which strongly overlapped
the smaller peak of mannose. For these cases, we kept reporting a sum value for Xyl/Man.

3.4 DFCHO and CCHO in saline field samples from different regions.
Several real samples were analyzed on DFCHO and CCHO (**Table 5** and **6**). In both sugar fractions, glucose
was the most abundant monosaccharide, as it has been reported before (Panagiotopoulos and Sempéré,
2005). In some of the samples, free fructose could be determined reaching concentrations comparable to
glucose. However, fructose cannot be determined in CCHO, since hydrolysis leads to complete destruction
of this sugar. High DFCHO was found in the samples from the Arctic including brine and ice core samples
reaching up to 118 nM in comparison to seawater samples from the Atlantic SWS 1-4 (11-15 nM).
However, a clear regional trend could not be identified for CCHO with concentrations, ranging between
260-1410 nM. Traces of free amino sugars and uronic acids were found after neutral desalinations.
However, a stronger enrichment is required in order to determine them quantitatively and will be the
focus of further studies.

























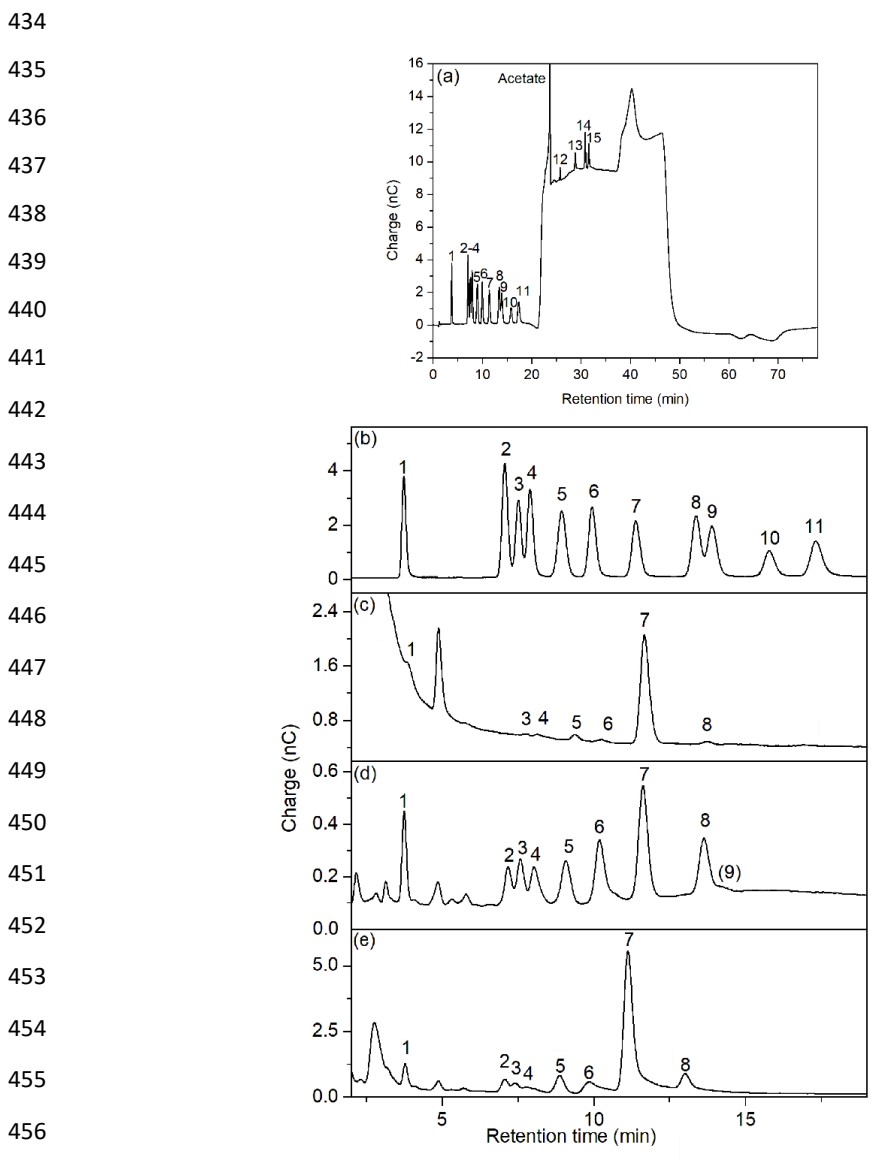

**Figure 5** Chromatograms of a) full chromatogram of standard solution 100 µg·L⁻¹; b-e) neutral sugars and amino sugars of b) standard solution 100 µg·L⁻¹; c) DFCHO in brine (B2) desalinated at natural pH; d) CCHO in a seawater sample (SWS 3); e) CCHO in Arctic brine (B2). 1 fructose, 2 galactosamine, 3 rhamnose, 4 arabinose, 5 glucosamine, 6 galactose, 7 glucose, 8 xylose, 9 mannose, 10 fructose, 11 ribose, 12 muramic acid, 13 galacturonic acid, 14 glucuronic acid, 15 mannuronic acid.







**Table 5** Mol percentages of individual neutral monosaccharides within DFCHO in seawater, Arctic brine and ice core samples.
<LOD stands for below detection limit.

|  | Glc mol% | Gal mol% | Xyl/Man mol% | Rha mol% | Fuc mol% | Ara mol% | Fru mol% | Total DFCHO nM |
|---|---|---|---|---|---|---|---|---|
| SWS 1 | 57 | 8 | <LOD | <LOD | 7 | 8 | 20 | 15 |
| SWS 2 | 52 | <LOD | <LOD | <LOD | <LOD | <LOD | 48 | 14 |
| SWS 3 | 80 | <LOD | 15 | <LOD | <LOD | 5 | <LOD | 11 |
| SWS 4 | 87 | <LOD | 5 | <LOD | <LOD | 8 | <LOD | 14 |
| SWS 5 | 89 | 3 | 3 | <LOD | <LOD | 4 | <LOD | 35 |
| SWS 6 | 16 | 8 | 22 | <LOD | 34 | 20 | <LOD | 27 |
| IC 1 | 50 | 6 | 2 | 3 | 15 | <LOD | 25 | 118 |
| B 1 | 23 | 27 | 13 | 8 | 29 | <LOD | <LOD | 15 |
| B 2 | 85 | 2 | 3 | 2 | 6 | 1 | <LOD | 53 |



**Table 6** Mol percentages of individual monosaccharides within CCHO in seawater, Arctic brine and ice core samples. <LOD
stands for below detection limit.

|  | Glc mol% | Gal mol% | Xyl/Man mol% | Rha mol% | Fuc mol% | Ara mol% | GalN mol% | GluN mol% | Gal-ac mol% | Gluc-ac mol% | Total CCHO nM |
|---|---|---|---|---|---|---|---|---|---|---|---|
| SWS 1 | 20 | 15 | 22 | 7 | 7 | 8 | 3 | 4 | 12 | 3 | 680 |
| SWS 2 | 25 | 12 | 24 | 8 | 7 | 7 | 2 | 3 | 11 | 1 | 290 |
| SWS 3 | 20 | 14 | 21 | 9 | 10 | 9 | 3 | 6 | 7 | 2 | 580 |
| SWS 4 | 31 | 12 | 26 | 5 | 6 | 7 | 2 | 3 | 6 | 2 | 410 |
| SWS 5 | 84 | 3 | 2 | 2 | 2 | 2 | 1 | 1 | <LOD | 3 | 1410 |
| SWS 6 | 48 | 7 | 10 | 4 | 2 | <LOD | 2 | 4 | 15 | 7 | 260 |
| IC 1 | 54 | 9 | 11 | 1 | <LOD | 1 | 2 | 1 | 10 | 10 | 330 |
| B 1 | 47 | 11 | 13 | 4 | 4 | 3 | 3 | 5 | <LOD | 10 | 420 |
| B 2 | 65 | 8 | 10 | 3 | 4 | 1 | 2 | 3 | <LOD | 3 | 640 |



## 5. Summary and conclusion
In this study, a novel protocol was presented for the analysis of both DFCHO and CCHO in saline aqueous
samples by applying HPACE-PAD with prior desalination by electro-dialysis. Recovery rates for neutral
monosaccharides ranged between 95-98%. By adjusting pH, charged monosaccharides such as free amino
sugars and uronic acids could be recovered with 58-59% at pH = 11 and 45-49% at pH = 1.5, respectively.
Dissolved polysaccharide standards, such as laminarin and alginic acid showed good recovery rates of 91-
93%, while a suspension of insoluble cellulose was quite difficult to recover reproducibly. Hence, electro-
dialysis for carbohydrate analysis is recommended to be used for filtered samples or for samples with low



amount of particulate matter. In this study, the osmotic and electro-osmotic loss of water was considered
in order to avoid an overestimation of the determined concentrations. In real seawater from different
locations, Arctic brine and sea ice core samples, CCHO was found in concentrations between 260 and
1410 nM. DFCHO ranged in much lower concentrations with 11-118 nM. Within both, DFCHO and CCHO,
the most dominant monosaccharide was glucose, followed by other neutral sugars.
In this study, the successful application of electro-dialysis in combination with HPAEC-PAD for the analysis
of marine carbohydrates (both free and combined) in marine matrices, such as seawater, ice cores and
brine could be demonstrated. The application of electro-dialysis for other more salt sensitive analyses
should be the focus of further researches, e.g. the reported interference of suspended sea spray aerosol
in Arctic snow samples during the quantification of insoluble light absorbing impurities such as black
carbon and dust performed via nebulization. Hence, this developed method has the potential to contribute
strongly in further research studies understanding biogeochemical processes in the oceans and related
saline matrices and sea-air exchange processes, especially for studying hot spot regions of climate change,
such as the Arctic.

## Acknowledgements



We gratefully acknowledge the funding by the Deutsche Forschungsgemeinschaft (DFG, German Research
Foundation) – Project-ID 268020496 – TRR 172, within the Transregional Collaborative Research Center
"ArctiC Amplification: Climate Relevant Atmospheric and SurfaCe Processes, and Feedback Mechanisms
(AC)³"in sub-projects B04. Additional support through the Leibniz Association SAW funding of the project
"Marine biological production, organic aerosol particles and marine clouds: a Process Chain
(MarParCloud)", (SAW-2016-TROPOS-2) is also gratefully acknowledged. We thank Svantje Pöge for
supporting work in the laboratory. We thank Jon Roa for CHO analysis following the Engel & Händel (2011)
protocol. We thank Susanne Fuchs, Kristin Recklies and Christian Weller of Thermo Fisher in his in-house
collaborations for fruitful discussions. We thank Patrick Altmeier and Philip Kalkhoff from PCCell GmbH
(Heusweiler, Germany) for guidance and technical support for helping to put electro-dialysis to operation
in our laboratory.



*Data availability.* The data will be available through the World Data Center PANGAEA (https://www.pangaea.de/) in the near future.

*Author contributions.* SZ wrote the manuscript with contributions from MvP, HH and AE. SZ and MvP collected seawater samples during different field campaigns. SZ optimized the presented method and performed the chemical measurements. AE performed CCHO analysis with supplied samples following her published protocol for evaluation purposes. All co-authors proofread and commented the manuscript.

*Competing interest.* The authors declare that they have no conflict of interest.

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
