# Peer review of "A protocol for quantifying mono- and polysaccharides in seawater and related saline matrices by electro-dialysis (ED) – combined with HPAEC-PAD"

_Ocean Science, 2020_

## Referee Comment (RC1) · Anonymous Referee #1 · 24 Feb 2020

**Review of manuscript os-2020-2 - A protocol for quantifying mono- and polysaccharides in seawater and related saline matrices by electro-dialysis (ED) – combined to HPAEC-PAD**

The manuscript discusses a novel way of desalinating marine samples for the determination of several low concentration organic compounds in that environment. The application of electrodialysis is investigated methodically, including the optimization of operational parameters and quantification of biases as well as a comparison to membrane dialysis. This work is very viable to help elucidate the composition and concentration of organics in marine samples and beyond. However, the text is not always easy and clear to read, and some structural changes and clarifications are needed before publication. These are discussed in the comments below.

**Specific comments**

- Lines 102-105: what are you basing this statement on? Is this based on preliminary own experiments? If so, can this be discussed further (in supplementary information perhaps)?
- Line 123: at which concentration was the seawater prepared?
- Line 149: why did you chose to work with a 16 g/L NaCl solution in the concentrate? The unit of ml/mL also seems wrong here.
- Line 155-156: what do you mean by 'homogenized with a pipette during desalination'? This is not clear to me.
- What type of membranes were the end membranes? From Figure 1 it seems like a CEM was used at the anode side and an AEM was used at the cathode side, but this is not specified in the text.
- Line 221: what are the setpoints 25V and 0.6A based on? This is a very high voltage which can cause water splitting. Did you see any pH fluctuations? This question is later answered in part 3.1, I suggest to already make a reference to this part and/or include the protocol for the parameter optimization in M&M instead of results.
- The same comments holds for part 3.2. Both this part and the previous part contains information that is not considered results or discussion and should thus be included in the introduction part (e.g. general explanation of (electro)osmotic water transport and why a concentration of 16 g/L was chosen in the concentrate).
- Line 319: it is not clear how you estimated this 3% and how you distinguished this osmotic water transport from the overwhelming electro-osmotic water transport. Is this from Figure 3? Because you can't really distinguish between the two modes of water transport during the first part of your desalination. The contribution of osmosis also changes in size and direction throughout the experiment as the salt concentrations change. Is it not simply enough to determine the final volumes in each compartment to account for concentration/dilution of your sample due to water transport?
- Line 351-352: transport of organics in presence of high salt concentrations is expected to be minimal, as demonstrated in a paper by Vanoppen et al. (2015). DOI 10.1021/es504389q
- Line 366-368: this statement is odd here and would be expected more at the end of the introduction.
- Figure 4: describe the difference between the full and dotted line in the caption. Please discuss the implications of the dotted line in the discussion. Is this a good quantification of the difference between both methods?

**Technical corrections**

- Generally, I propose to introduce the abbreviation ED for electro-dialysis and using it throughout the manuscript.
- Line 98, replace ',' with 'and' (and there are ofcourse many more examples of ED application)
- Line 199: sometimes you use 'electro-dialysis' and sometimes 'electrodialysis'. The latter is more frequently used and you can introduce the abbreviation as suggested before.
- Line 210: 'slide' = 'slight'

---

## Referee Comment (RC2) · Anonymous Referee #2 · 28 Apr 2020

This is a detailed and thorough analytical development paper applied to a number of matrices and tested using marine samples. The authors have managed to achieve sensitive detection limits for a challenging analysis and the paper is suitable for publication with minor revision. I have detailed the changes needed below:

DFCHO and CCHO are not obvious abbreviations; are these accepted forms? L13. 'dissolved free'; should also be DCCHO in that case. L20. Delete 'real'. L45. 'with' not 'to'. L50. 'recent' not 'latest'. L57. Analogous to DFCC and DCAA? L68. 'oceanic environments' is more appropriate. L74. kinds L75. gas chromatography L76. How is

it labour intensive; give brief details? L81. The 'high ionic strength/content of seawater samples' is better. L107. Related saline samples; what are they? L116. Resistivity, not conductivity. L117. How long were items soaked in 10 % HCl? L123. 'from' not 'to' L129. Delete 'real' L131. Add 'sampling campaigns; delete 'of our department' and add any details to acknowledgements. L132. Delete 'kept'. L143. mL ; change throughout. L149. I presume this is 60 mL.min-1 ; space before 'Two' L150. 'compartment' or 'section' 'containing' the electrodes. L152. 'made of' stainless steel. L153. (e.g. to end of sentence) L155-156. Explain more clearly how homogenisation was achieved. L156. Renewed how often (based on number of samples?)? L163. 'filled with' or simiar L171. Did the guard and analytical columns have the same packing (different codes given)? L172. What was maintained at 30 oC, and how? L173. Adaptation of Meyer et al. (2008) L174. 'were eluted in 4 nM NaOH solution'. L175. Were they contaminants? L176. 'the remaining . . . . Equilibrated with 4 mM NaOH solution. L179. 'in' not 'as a' L180. 'ranged from 2-12 nM L181. with reported data (refs) L183. resistivity < 18.2. . .. L193. Do you know how the pH changed with each change in the gradient profile? L198. 4 oC; insert space between numbers and units through the paper. L199. 'at the end'. L202. 'of expected DFCHO concentrations in seawater'. L204. Weighed; change throughout. L205. Delete 'remaining'. L208. 'in' duplicate L223. solutions L224. 'repeated in triplicate for four. . . ..'; delete 'and as triplicate for each time'. L234. Replace 'as well with' by 'and'; rmove comma after membranes. L248. The samples can't be neutralised by evaporation; clarify this text. L259-260. 'requires prior removal of sea salt'. L283. 'Large pH increase' L301. What is hydrated water; is it the hydronium ion? L330. 'of 87 %' L339. 'a constant rate' L342. 'at the end' L366. '89 % recovered at pH 1.5' L381. 'it does not leave' L383. Replace 'worse' with 'lower'. L387. Replace 'gadget' with 'system' or 'apparatus'. L392. 'to filtered samples' L396. 'were performed' L398. 'method presented here' L416. 'been reported'; delete 'given only' L479. 'of' not 'with' L484. research

---

## Author Comment (AC1) · 26 May 2020

**Answers to the Referees' comments regarding the manuscript:**

**A protocol for quantifying mono- and polysaccharides in seawater and related saline matrices**
**by electro-dialysis (ED) – combined with HPAEC-PAD**

**Sebastian Zeppenfeld[1], Manuela van Pinxteren[1], Anja Engel[2], Hartmut Herrmann[1,*]**

1 Atmospheric Chemistry Department (ACD), Leibniz-Institute for Tropospheric Research (TROPOS),
Leipzig, Germany

2 GEOMAR Helmholtz Centre for Ocean Research Kiel, Kiel, Germany

*Correspondence to*: Hartmut Herrmann (herrmann@tropos.de)

Submitted to Ocean Science, ACP/OS special issue: 'Marine organic matter: from biological production in
the ocean to organic aerosol particles and marine clouds'

Manuscript ID: os-2020-2

We thank both reviewers for the evaluation of our manuscript. In this document, all of their constructive
comments were answered thoroughly. The referees' comments are marked blue and our replies black.
The given line numbers of changed sentences are referring to the new lines in the revised manuscript.

 **Reviewer: 2, 28 Apr 2020**

*This is a detailed and thorough analytical development paper applied to a number of matrices and tested*
*using marine samples. The authors have managed to achieve sensitive detection limits for a challenging*
*analysis and the paper is suitable for publication with minor revision. I have detailed the changes needed*
*below:*

Authors: Thank you. Based on your very useful comments, we performed following changes in the
manuscript.

*DFCHO and CCHO are not obvious abbreviations; are these accepted forms?*

Authors: We agree that DFCHO and CCHO are not obvious abbreviations. We assume that CHO is originally
derived from the aldehyde group as an important structural element of carbohydrates. However, among
others, these abbreviations are frequently used within the marine chemistry community (e.g. in Borchard
and Engel, 2015; Engel and Händel, 2011; Jugnia et al., 2006; Richardot et al., 1999; Tranvik and Jørgensen,
1995). Therefore, we prefer to keep these abbreviations.

*L13.'dissolved free'; should also be DCCHO in that case.*

Authors: We replaced 'free (DFCHO) and combined monosaccharides (CCHO)' with 'dissolved free (DFCHO)
and dissolved combined carbohydrates (DCCHO)' (new lines 13-14). Furthermore, we replaced 'CCHO' with
'DCCHO' throughout the manuscript. Additionally, we added the sentence, which reads: 'In aquatic
environments, CCHO either appear in a particulate (PCCHO) or dissolved form (DCCHO).' (new lines 40-41)
Furthermore, we replaced 'Rather, we recommend ED only for the application to filtered samples
(dissolved compounds), while particulate organic matter might be better analyzed from filters after
filtration.' with 'Rather, we recommend ED only for the application to filtered samples (DCCHO), while
PCCHO might be better analyzed from filters after filtration.' (new lines 392-393)

*L20. Delete 'real'.*

Authors: We deleted 'real'. The new sentence now reads: 'The applicability of this method for the analysis
of DCCHO was evaluated with standard solution and seawater samples compared with another established
desalination method using membrane dialysis.' (new lines 19-21)

*L45. 'with'not 'to'.*

Authors: The word 'to' was replaced with 'with'. The new sentence now reads: 'Furthermore, an elevated
release of polysaccharides by phytoplankton, mostly of gelatinous nature, has been associated with stress
situations, such as a deficiency of nutrients, freezing or fluctuating water potential….' (new lines 45-47)

*L50. 'recent' not 'latest'.*

Authors: We replaced 'a latest study' with 'a recent study'.

**L57. Analogous to DFCC and DCAA?**

Authors: We believe that the reviewer was referring to DFAA (dissolved free amino acids) and DCAA (dissolved combined amino acids). It is true that DFAA and DFCHO in seawater are mostly found in lower concentrations than their macromolecular equivalents (DCAA/DCCHO). Previous publication explained this finding with marine microbes processing these free sugars and amino acids with a very high turnover rate. We added this information to our manuscript, which now reads: 'DFCHO are mostly found in lower concentrations than DCCHO, since marine microbes utilize them with high turnover rates (Engbrodt, 2001; Engel and Händel, 2011; Ittekkot et al., 1981; Thornton et al., 2016) as it has been reported for amino acids analogously as well (Kuznetsova and Lee, 2002).' (new lines 57-59)

**L68. 'oceanicenvironments' is more appropriate.**

Authors: We replaced 'maritime regions' with 'marine environments'. (new line 68)

**L74. kinds**

Authors: We replaced 'with different kind of chromatographic methods' with 'with 'different kinds of chromatographic methods'. (new line 74)

**L75. gas chromatography**

Authors: We replaced 'gas chromatograph' with 'gas chromatography'. (new line 75)

**L76. How is it labour intensive; give brief details?**

Authors: There are several ways to derivatize sugars depending on the applied chromatographic analysis, requiring the use of toxic chemicals, robust lab parameters and internal standards. Derivatization is not needed when HPAEC-PAD is applied. However, we came to the conclusion that our use of the word 'labour intensive' is our subjective opinion and possibly misleading. Since this word is not important for understanding the text, we decided to delete it and rephrase the sentence. We replaced 'These methods require a quite difficult sample preparation, including a labor intensive derivatization step' with 'These methods require a prior derivatization in order to enable the chromatographic separation and detectability of these carbohydrates (Panagiotopoulos and Sempéré, 2005)'. (new lines 75-77)

**L81. The 'high ionic strength/content of seawatersamples' is better.**

Authors: We replaced 'the presence of sea salt in seawater samples' with 'the high ionic content in seawater samples' (new line 81)

**L107. Related saline samples; what are they?**

Authors: We agree that the term 'related saline samples' is not precise. For being more concrete, we added the examples ice cores and brine from Arctic sea ice. The new sentence now reads: 'Within the present study, a novel protocol for the desalination of seawater samples and other saline samples (e.g. ice cores and brine from Arctic sea ice), applying electro-dialysis and HPAEC-PAD is presented, accounting for the described biases.' (new line 115-117)

**L116. Resistivity, not conductivity.**

Authors: We replaced 'conductivity' with 'resistivity'. (new line 127)

**L117. How long were items soaked in 10 % HCl?**

Authors: The plastic items were rinsed with 10% HCl three times. We added this information to the main text, which now reads: 'All plastic equipment was first rinsed with 10% HCl solution for three times and then washed with ultrapure water another three times.' (new lines 128-129)

**L123. 'from' not 'to'**

Authors: We replaced 'Synthetic seawater samples were made of commercially available sea salts (Sigma)' with 'Synthetic seawater samples were made from commercially available sea salts (Sigma)'. (new lines 134-135)

**L129. Delete 'real'**

Authors: Done. We applied this change throughout the manuscript.

**L131. Add 'sampling campaigns; delete'of our department'**

Authors: We changed this sentence, which now reads: 'These saline samples were collected during different sampling campaigns and stored at -20 °C.' (new lines 147-148)

**and add any details to acknowledgements.**

Authors: Additional details about the sampling campaigns, such as locations and dates, are given in Table 1. Furthermore, we added a sentence to the acknowledgments: 'We thank for the opportunities to use aqueous samples from various sampling campaigns in order to develop the method presented here.' (new lines 497-498)

**L132. Delete 'kept'.**

Authors: Done. The new sentence now reads: 'These saline samples were collected during different sampling campaigns and stored at -20 °C' (new lines 147-148)

**L143. mL ; change throughout.**

Authors: We replaced 'ml' with 'mL' throughout the manuscript. Furthermore, we replaced 'µl' with 'µL'
throughout the manuscript.

**L149. I presume this is 60 mL.min-1 ; space before 'Two'**

Authors: Yes, thank you. This was a typing mistake. The new sentence now reads: 'This solution was
circulated at a rate of 60 mL·min$^{-1}$. Two end…'(new lines 163-164)

**L150. 'compartment' or'section' 'containing' the electrodes.**

Authors: We replaced 'the third department including the electrodes' with 'the third compartment
containing the electrodes'. (new line 165)

**L152. 'made of' stainless steel.**

Authors: We replaced the word 'based on' with 'made of'. The new sentence reads: 'The MMO cathode
was made of stainless steel.' (new lines 168-169)

**L153. (e.g. toend of sentence)**

We replaced 'for avoiding unwanted redox reactions, e.g. the generation of corrosive elemental chlorine
from chloride.' with 'for avoiding unwanted redox reactions (e.g. the generation of corrosive elemental
chlorine from chloride).' (new lines 170-171)

**L155-156. Explain more clearly how homogenisation was achieved.**

Authors: In order to describe more clearly how homogenization was achieved, we rephrased the sentence,
which now reads: 'The sample solution was homogenized during each desalination by drawing some liquid
into a Pasteur pipette and draining it immediately back to the sample compartment.' (new lines 172-173)

**L156. Renewed how often (based on number of samples?)?**

Authors: We renewed these solutions after every tenth desalination. The new text now reads: 'The
electrolyte and the concentration solutions were renewed after every tenth desalination.' (new line 174)

**L163. 'filled with' or simiar**

Authors: We replaced 'exposed to' with 'filled with'. (new line 182)

**L171. Did the guard and analytical columns have the same packing (different codesgiven)?**

Authors: To our knowledge, the guard and analytical column do have the same packing. The only difference
between these both columns is their length. Therefore, the given code for both columns is almost identical with 'Dionex CarboPac PA20 analytical column (3x150mm)' and 'Dionex CarboPac PA20 guard column
(3x30mm)'. However, we missed writing 'PA' in 'Dionex CarboPac PA20 analytical column (3x150mm)'.
This was corrected now.

**L172. What was maintained at 30 oC, and how?**

Authors: The analytical column and guard column were permanently maintained at 30 °C by keeping them
in a column oven. In order to make this clearer to the reader, we rephrased the sentence, which now
reads: 'Several neutral monosaccharides, amino sugars and uronic acids were separated on a Dionex
CarboPac PA20 analytical column (3x150mm) combined with a Dionex CarboPac PA20 guard column
(3x30mm). The column oven temperature was maintained at 30 °C.' (new lines 189-191)

**L173. Adaptation of Meyer etal. (2008)**

Authors: We replaced 'an adaption to the elution by (Meyer et al., 2008).' with 'an adaption of Meyer et
al. (2008).' (new lines 192-193)

**L174. 'were eluted in 4 nM NaOH solution'.**

Authors: We rephrased the sentence, which now reads: 'Neutral and amino sugars were eluted in 4 mM
NaOH within the first 19 min.' (new line 193)

**L175. Were they contaminants?**

Authors: Sugar acids are not contaminants, but interesting analytes that we want to quantify. These sugar
acids elute from the analytical column when sodium acetate is added to the eluent, since they interact
strongly with the stationary phase. At the same time, contaminants are flushed from the column as well,
when sodium acetate is added. In order to improve the understandability to the reader, we rephrased the
sentence, which now reads: 'By adding sodium acetate, sugar acids eluted. At the same time, organic and
inorganic contaminants were flushed from the column.' (new lines 193-195)

**L176. 'the remaining.... Equilibrated with 4 mM NaOH solution.**

Authors: We added the word 'the', and replaced 'at' with' with'. The sentence now reads: 'After the
removal of the remaining acetate by 250 mM NaOH, the system was equilibrated with 4 mM NaOH for the
next sample injection.' (new lines 195-196)

**L179. 'in' not 'asa'**

Authors: 'As a duplicate' was replaced with 'in duplicate'. Furthermore, we replaced 'as triplicate' with 'in
triplicate' throughout the manuscript.

**L180. 'ranged from 2-12 nM**

Authors: We replaced 'were ranging between 2-12 nM' with 'ranged from 2-12 nM'. (new line 199)

**L181. with reported data (refs)**

Authors: We replaced 'in good agreement with literature' with 'in good agreement with reported data'.
(new line 200)

**L183. resistivity <18.2....**

Authors: We changed 'conductivity' to 'resistivity'. (new line 202) Thank you for pointing on this oversight.

**L193. Do you know how the pH changed with each change in the gradientprofile?**

Authors: An integrated pH reference electrode measures the pH, which is displayed online. We observed
a constant pH of 12.0 from 0 min to 19 min. By adding eluent C from 19 min to 35 min, the pH continuously
raised until reaching a pH=13. Setting eluent A on 100% from 35 min to 44 min resulted into a permanent
increase of pH until 13.5. Setting all eluents on their initial concentrations caused a slow adaption to
pH=12.0 from 44 min to 78 min for the next injection. However, we did not add this information to the
manuscript, since we don't believe that it has an important significance for the paper

**L198. 4 oC; insert space between numbers and units through the paper.**

Authors: We inserted a space between numbers and '°C' throughout the manuscript.

**L199. 'at the end'.**

Authors: We changed 'in the end' to 'at the end' throughout the manuscript.

**L202. 'of expected DFCHO concentrations in seawater'.**

Authors: We replaced 'A concentration step using a vacuum concentrator (MiVac) at 55 °C allowed the
detection of low concentrated DFCHO, as it occurs in most seawater samples.' with 'A concentration step
using a vacuum concentrator (MiVac) at 55 °C allowed the detection of expected DFCHO concentrations
in seawater.' (new lines 219-220)

**L204.Weighed; change throughout.**

Authors: 'Weighted' was replaced with 'weighed' throughout the manuscript.

**L205. Delete 'remaining'.**

Authors: Done. The changed sentence now reads: 'After reaching a volume of less than approximately
600 µl,…'.(new lines 221-222)

**L208. 'in' duplicate**

Authors: 'as duplicate' was replaced with 'in duplicate' throughout the manuscript.

**L223.solutions**

Authors: 'solution' was replaced with 'solutions'.(new line 239)

**L224. 'repeated in triplicate for four…..'; delete 'and as triplicate for eachtime'.**

Authors: We rephrased this sentence, which now reads: 'These measurements were repeated in triplicate
for four different sea salt solutions (10, 20, 30 and 40 PSU).'(new lines 239-240)

**L234. Replace 'as well with' by 'and'; rmove comma after membrances.**

Authors: We replaced 'as well with' with 'and'. We removed the comma after 'membranes. The revised
sentence now reads: 'In order to account for possible wasting phenomena, repetitions were performed
with new membranes and membranes which already had been used for some time before.' (new lines
249-251)

**L248.The samples can't be neutralised by evaporation; clarify this text.**

Authors: One crucial step for the sample treatment is the neutralization of the sample after acid hydrolysis.
However, the neutralization of acids by the addition of a base (e.g. NaOH) will introduce new ions to the
sample, which disturb the analysis at HPAEC-PAD. Hence, a neutralization using a base appears quite
pointless after a prior desalination.

The advantage of using hydrochloric acid is the volatility of HCl, when the contained water molecules
evaporate at the same time. By removing HCl from the system by evaporation, a neutralization can actually
be achieved. Amongst other references, this procedure has been already described by Engel and Händel
(2011) and Panagiotopoulos and Sempéré (2005).

In order to make this approach clearer to the reader, we rephrased the sentence, which now reads:
'Aliquots of 1 mL with and without desalinations were hydrolyzed (HCl 0.8 M, 100 °C, 20 h) and neutralized
by evaporation of the volatile liquid..' (new lines 262-263)

**L259-260. 'requiresprior removal of sea salt'.**

Authors: We removed 'requires a prior removal of disturbing sea salt.' with 'requires prior removal of sea salt.' (new lines 275-276)

**L283. 'Large pH increase'**

Authors: We replaced 'strong rise of the pH' with 'large pH increase'. (new line 298)

**L301. What is hydrated water; isit the hydronium ion?**

Authors: We actually meant neutral water, which is bound to ions in their hydration sphere. We corrected the sentence, which now reads: 'By operating an electrical field, the active transport of charged molecules (*migration*) and water bound to ions in a hydration sphere (*electro-osmosis*) takes place…' (new lines 107-108)

**L330. 'of 87 %'**

Authors: We agree that the used preposition 'onto 87 %' is wrong. In order to give a unmistakable phrasing, we changed the sentence which now reads: 'a maximal reduction of the sample volume by 13 % due to electro-osmosis was expected' (new lines 333-334)

**L339. 'a constant rate'**

Authors: We added the word 'a'. The new sentence now reads: 'Once the sea salt is removed, osmotic water transfer remains at a constant rate of approximately $0.1\% \cdot min^{-1}$. (new lines 342-343)

**L342. 'at the end'**

Authors: We replaced 'in the end' with 'at the end'. (new line 346)

**L366.'89 % recovered at pH 1.5'**

Authors: We replaced '…with the exception of fructose, which was recovered with 89% at pH 1.5,…' with '…with the exception of fructose, which was 89% recovered at pH 1.5,…'. (new lines 371-372)

**L381. 'it does not leave'**

Authors: We added the word 'it'. The new sentence now reads: '…and it does not leave the sample solution'. (new line 382)

**L383. Replace 'worse' with 'lower'.**

Authors: We replaced 'in much worse recoveries' with 'in much lower recoveries'. (new line 384)

**L387. Replace 'gadget' with 'system' or 'apparatus'.**

Authors: We replaced 'gadget' with 'apparatus'. (new line 387)

**L392. 'to filtered samples'**

Authors: We replaced 'at filtered samples' with 'to filtered samples'. (new line 392)

**L396.'were performed'**

Authors: We replaced 'studies have been performed' with 'studies were performed'. (new lines 396-397)

**L398. 'method presented here'**

Authors: We replaced 'the here presented method' with 'the method presented here'. (new line 398)

**L416. 'been reported'; delete 'givenonly'**

Authors: We deleted the word 'given only and added 'been reported'. The changed sentence now reads: 'Therefore, xylose and mannose have been reported as sum concentrations frequently.' (new line 416-417)

**L479. 'of' not 'with'**

Authors: We replaced 'lower concentrations with 11-118 nM' with 'lower concentrations of 11-118 nM'. (new line 479)

**L484. research**

Authors: We replaced 'further researches' with 'further research'. (new line 486)

**Additional changes**

We replaced 'combined to' with 'combined with' (title).

We added 'hexoses, pentoses' to line 39.

**Cited references:**

Borchard, C. and Engel, A.: Size-fractionated dissolved primary production and carbohydrate composition
of the coccolithophore *Emiliania huxleyi*, Biogeosciences, 12(4), 1271–1284, doi:10.5194/bg-12-1271-
2015, 2015.

Engbrodt, R.: Biogeochemistry of dissolved carbohydrates in the Arctic, Berichte zur Polar-und
Meeresforschung (Reports on Polar and Marine Research), 396, 106pp, 2001.

Engel, A. and Händel, N.: A novel protocol for determining the concentration and composition of sugars
in particulate and in high molecular weight dissolved organic matter (HMW-DOM) in seawater, Marine
Chemistry, 127(1), 180–191, doi:10.1016/j.marchem.2011.09.004, 2011.

Ittekkot, V., Brockmann, U., Michaelis, W. and Degens, E. T.: Dissolved free and combined carbohydrates
during a phytoplankton bloom in the northern North Sea, Marine Ecology Progress Series, 4, 299–305,
1981.

Jugnia, L.-B., Richardot, M., Debroas, D. and Dévaux, J.: Bacterial Production in the Recently Flooded Sep
Reservoir: Diel Changes in Relation to Dissolved Carbohydrates and Combined Amino Acids,
Hydrobiologia, 563(1), 421–430, doi:10.1007/s10750-006-0039-x, 2006.

Kuznetsova, M. and Lee, C.: Dissolved free and combined amino acids in nearshore seawater, sea surface
microlayers and foams: Influence of extracellular hydrolysis, Aquatic sciences, 64(3), 252–268, 2002.

Meyer, A., Fischer, H., Kuzyakov, Y. and Fischer, K.: Improved RP-HPLC and anion-exchange
chromatography methods for the determination of amino acids and carbohydrates in soil solutions, J.
Plant Nutr. Soil Sci., 171(6), 917–926, doi:10.1002/jpln.200700235, 2008.

Panagiotopoulos, C. and Sempéré, R.: Analytical methods for the determination of sugars in marine
samples: A historical perspective and future directions, Limnology and Oceanography: Methods, 3(10),
419–454, doi:10.4319/lom.2005.3.419, 2005.

Richardot, M., Debroas, D., Thouvenot, A., Romagoux, J. C., Berthon, J. L. and Devaux, J.: Proteolytic and
glycolytic activities in size-fractionated surface water samples from an oligotrophic reservoir in relation
to plankton communities, Aquat. sci., 61(4), 279–292, doi:10.1007/s000270050066, 1999.

Thornton, D. C. O., Brooks, S. D. and Chen, J.: Protein and Carbohydrate Exopolymer Particles in the Sea
Surface Microlayer (SML), Front. Mar. Sci., 3, 135–143, doi:10.3389/fmars.2016.00135, 2016.

Tranvik, L. J. and Jørgensen, N. O. G.: Colloidal and dissolved organic matter in lake water: Carbohydrate
and amino acid composition, and ability to support bacterial growth, Biogeochemistry, 30(2), 77–97,
doi:10.1007/BF00002725, 1995.

---

## Author Comment (AC2) · 26 May 2020

| 1 | Answers to the Peferees' | commonts regarding | the manuscript.   |
|---|--------------------------|--------------------|-------------------|
| T | Answers to the reletees  | comments regarding | s the manuscript. |

| 3
4         | A protocol for quantifying mono- and polysaccharides in seawater and related saline matrices by electro-dialysis (ED) – combined with HPAEC-PAD                                                                                                                                                                  |  |
|----------------|------------------------------------------------------------------------------------------------------------------------------------------------------------------------------------------------------------------------------------------------------------------------------------------------------------------|--|
| 5              | Sebastian Zeppenfeld 1 , Manuela van Pinxteren 1 , Anja Engel 2 , Hartmut Herrmann 1,*                                                                                                                                                                               |  |
| 6              |                                                                                                                                                                                                                                                                                                                  |  |
| 7
8         | 1 Atmospheric Chemistry Department (ACD), Leibniz-Institute for Tropospheric Research (TROPOS), Leipzig, Germany                                                                                                                                                                                                 |  |
| 9              | 2 GEOMAR Helmholtz Centre for Ocean Research Kiel, Kiel, Germany                                                                                                                                                                                                                                                 |  |
| 10             | *Correspondence to: Hartmut Herrmann (herrmann@tropos.de)                                                                                                                                                                                                                                                        |  |
| 11             |                                                                                                                                                                                                                                                                                                                  |  |
| 12
13       | Submitted to Ocean Science, ACP/OS special issue: 'Marine organic matter: from biological production in the ocean to organic aerosol particles and marine clouds'                                                                                                                                                |  |
| 14             |                                                                                                                                                                                                                                                                                                                  |  |
| 15             | Manuscript ID: os-2020-2                                                                                                                                                                                                                                                                                         |  |
| 16             |                                                                                                                                                                                                                                                                                                                  |  |
| 17             |                                                                                                                                                                                                                                                                                                                  |  |
| 18
19
20 | We thank both reviewers for the evaluation of our manuscript. In this document, all of their constructive comments were answered thoroughly. The referees' comments are marked blue and our replies black. The given line numbers of changed sentences are referring to the new lines in the revised manuscript. |  |
| 21             |                                                                                                                                                                                                                                                                                                                  |  |
| 22             |                                                                                                                                                                                                                                                                                                                  |  |
| 23             |                                                                                                                                                                                                                                                                                                                  |  |
| 24             |                                                                                                                                                                                                                                                                                                                  |  |
| 25             |                                                                                                                                                                                                                                                                                                                  |  |
| 26             |                                                                                                                                                                                                                                                                                                                  |  |
| 27             |                                                                                                                                                                                                                                                                                                                  |  |
| 28             |                                                                                                                                                                                                                                                                                                                  |  |
| 29             |                                                                                                                                                                                                                                                                                                                  |  |
| 30             |                                                                                                                                                                                                                                                                                                                  |  |
| 31             |                                                                                                                                                                                                                                                                                                                  |  |

**32 **Reviewer: 1, 24 Feb 2020**

33 The manuscript discusses a novel way of desalinating marine samples for the determination of several

34 low concentration organic compounds in that environment. The application of electrodialysis is

35 investigated methodically, including the optimization of operational parameters and quantification of

- 36 biases as well as a comparison to membrane dialysis. This work is very viable to help elucidate the
- 37 composition and concentration of organics in marine samples and beyond.
- However, the text is not always easy and clear to read, and some structural changes and clarifications
   are needed before publication. These are discussed in the comments below.
- 40 Authors: Thank you for your very constructive review. In order to improve the readability of our 41 manuscript, we carefully read your comments and changed the text correspondingly.
- 42

43 Specific comments:

- Lines 102-105: what are you basing this statement on? Is this based on preliminary own experiments? If so, can this be discussed further (in supplementary information perhaps)?
- 46 Authors: The mentioned phenomena (osmosis, electro-osmosis, water splitting, etc.) during electro-47 dialysis had been subject to previous publications (Galama et al., 2014; Han et al., 2015). However, the
- 48 impact of these phenomena on analytical guantifications has not been discussed yet in the literature. Here,
- 49 we explicitly include the discussion of such phenomena in chapter 3.1 and 3.2 of this manuscript regarding
- 50 the analysis of sugars in salty matrices. In order to make it clearer to the reader that these biases have
- 51 already been mentioned by previous studies before, we added further information to the introduction.
- 52 The changed text now reads: 'However, following biases, which have hitherto not been discussed in this
- analytical context, can occur during the application of ED and might falsify the determined concentration
- of the analytes in the sample. In contact with ion exchange membranes, the passive transport of water
- 55 (osmosis) and solutes with a low molecular weight (diffusion), such as DFCHO, can occur triggered by a
- 56 concentration gradient between the sample and concentration channels (Galama et al., 2014; Galier et al.,
- 57 2012). Additionally, the active transport of charged molecules (*migration*) and water bound to ions in their 58 hydration sphere (*electro-osmosis*) takes place by operating an electrical field (Galama et al., 2014; Han et
- 58 hydration sphere (*electro-osmosis*) takes place by operating an electrical field (Galama et al., 2014; Han et al., 2015, 2017). While osmosis and electro-osmosis induce an unavoidable loss of water and hence of the
- total volume of the sample, diffusion and migration of the analytes result in a loss of analyzable molecules.
- 61 Furthermore, water splitting and associated pH fluctuations have been reported, when a limiting current
- 62 is exceeded during an ED desalination (Cowan, 1962; Martí-Calatayud et al., 2018; Ottosen et al., 2000;
- 63 Vetter et al., 2007).' (new lines 102-113)
- 64

**65 Line 123: at which concentration was the seawater prepared?**

- 66 Authors: We used four different concentration resulting in salinities of 10, 20, 30 and 40 practical salinity 67 units (PSU). The information about the concentrations is given in chapter 2.6, where we explain the 68 concrete experimental set-up. Now we added this information to chapter 2.1 'Chemicals and materials' as 69 well. The changed sentence now reads: 'Synthetic seawater samples were made from commercially 70 available sea salts (Sigma) achieving four solutions with salinities of 10, 20, 30 and 40 practical salinity units 71 (PSU). The salinity and the pH of water aliquots was measured by using a conductivity meter (pH/Cond
- 72 3320, WTW).' (new lines 134-135) 73
- Line 149: why did you chose to work with a 16 g/L NaCl solution in the concentrate? The unit of ml/mL
   also seems wrong here.
- 76 Authors:
- 77 -16 g/L:
- A concentration of 16 g NaCL/L in the concentrate circuit was originally chosen in order to have a good
- 79 conductivity within the ED system and minimizing the impact of the osmotic transport, which could result

80 in a change of the DFCHO and DCCHO concentrations. Among other parameters, the effect of osmosis depends on the difference between the concentrations of solutes in the sample solution and the 81 82 concentration circuit ( $c_s - c_c$ ). In order to minimize the analytical error due to osmosis, we chose a 83 concentration which is approximately in the middle between the concentrations of a typical seawater 84 sample before (30-39 PSU) and after the desalination (0.2-0.4 PSU) for balancing the positive and negative 85 contribution of osmosis on the total sample volume during a typical desalination. We originally included 86 this issue in chapter 3.2 and think that this is a good place for this discussion. However, in the current 87 version, we added a short explanation in the 'Experimental', in order to explain the used concentration. The changed text now reads: 'For maintaining the conductivity within the system and receiving the sea 88 89 salt from the sample, the next compartment contained the concentration circuit, a 16 g·L-1 NaCl solution 90 (Merck). This concentration was chosen in order to minimize the osmotic water transfer as discussed 91 below.' (new lines 161-163) 92 93 -unit: 94 Thanks, it was a typing mistake. We changed the sentence, which now reads: 'This solution was circulated 95 at a rate of 60 mL·min-1.' (new line 164) 96 97 Line 155-156: what do you mean by 'homogenized with a pipette during desalination'? This is not clear 98 to me.-99 Authors: In order to describe more clearly how homogenization was achieved, we rephrased the sentence, 100 which now reads: 'The sample solution was homogenized during each desalination by drawing some liquid 101 into a Pasteur pipette and draining it immediately back to the sample compartment.' (new lines 172-173) 102 103 What type of membranes were the end membranes? 104 Authors: We added this information to the main text, which now reads: 'The end membranes were cation 105 exchange membranes with an increased chemical durability and an additional reinforcement in order to 106 withstand the strong differential pressure within the ED system.' (new lines 165-167) 107 108 From Figure 1 it seems like a CEM was used at the anode side and an AEM was used at the cathode side, 109 but this is not specified in the text. Authors: You are right. We specified this in the main text. The changed text now reads: 'The functionalized 110 111 anion exchange membrane (quaternary ammonium aliphatic polyether) and cation exchange membrane (sulfonated aromatic polyether) bordered this compartment on both sides. Depending on their chemical 112 113 properties, the membranes allowed exclusively the migration of either positively or negatively charged 114 ions. For that matter, the anion exchange membrane bordering the sample chamber was oriented to the 115 anode and the cation exchange membrane to the cathode.' (new lines 155-160) 116 117 Line 221: what are the set points 25V and 0.6A based on? This is a very high voltage which can cause 118 water splitting. Did you see any pH fluctuations? This question is later answered in part 3.1, I suggest to 119 already make a reference to this part and/or include the protocol for the parameter optimization in 120 M&M instead of results.-121 Authors:

122 - set points 25V and 0.6A:

123 The set points of maximal voltage and maximal current was based on technical information given by the

producer of the PCCell Micro Bench ED system. Furthermore, these parameters were adapted to our

application in order to achieve a fast desalination, but avoiding scaling of membranes caused by pH

- 126 fluctuations due to water splitting as it might occur during the exceedance of a limiting electrical current
- 127 (as it was already described by Vetter et al., 2007) We discussed this topic in chapter 3.1 of our manuscript.

- 128
- 129 high voltage can cause water splitting

130 To our knowledge, the occurrence of water splitting during an ED desalination is related to the applied

131 current and not directly to the voltage. E.g. Vetter et al. (2007) applied a maximal voltage of 250 V in their

- 132 ED device while caring about the applied current carefully. They did not observe pH fluctuations due to
- 133 water splitting.
- 134
- 135 pH fluctuations:

We observed strong pH fluctuations when the current was set too high during ED desalination. Wediscussed this phenomenon in chapter 3.1 of our manuscript.

- 138
- 139 adding reference to M&M:

140 In order to improve the understandability of the used ED parameters to the reader already in this part of 141 the manuscript, we added the information to the chapter '2.3 The ED system', where we mention the used 142 voltage and current for the first time. The added text reads: 'The maximal current was set on 0.6 A in order 143 to perform a fast desalination, but also to avoid a scaling of the membranes due to water splitting and is

- 144 discussed more in detail below.' (new lines 176-177)
- 145

The same comments holds for part 3.2. Both this part and the previous part contains information that is not considered results or discussion and should thus be included in the introduction part (e.g. general explanation of (electro)osmotic water transport and why a concentration of 16 g/L was chosen in the

- 149 concentrate).
- 150 Authors:
- 151 (Electro)-osmotic transport

152 According to the referees' suggestion, we restructured the manuscript by taking the general explanation 153 of (electro)osmotic water transport, diffusion and migration and water splitting processes from the results 154 into the introduction. The changed introduction now reads: 'However, following biases, which have 155 hitherto not been discussed in this analytical context, can occur during the application of ED and might 156 falsify the determined concentration of the analytes in the sample. In contact with ion exchange 157 membranes, the passive transport of water (osmosis) and solutes with a low molecular weight (diffusion), 158 such as DFCHO, can occur triggered by a concentration gradient between the sample and concentration 159 channels (Galama et al., 2014; Galier et al., 2012). Additionally, the active transport of charged molecules 160 (migration) and water bound to ions in their hydration sphere (*electro-osmosis*) takes place by operating 161 an electrical field (Galama et al., 2014; Han et al., 2015, 2017). While osmosis and electro-osmosis induce 162 an unavoidable loss of water and hence of the total volume of the sample, diffusion and migration of the 163 analytes result in a loss of analyzable molecules. Furthermore, water splitting and associated pH 164 fluctuations have been reported, when a limiting current is exceeded during an ED desalination (Cowan, 165 1962; Martí-Calatayud et al., 2018; Ottosen et al., 2000; Vetter et al., 2007).' (new lines 102-113)

166 - 16 g/L

167 As already mentioned before, we added a short explanation in the 'Experimental', in order to explain the 168 used concentration. The changed text now reads: 'For maintaining the conductivity within the system and 169 receiving the sea salt from the sample, the next compartment contained the concentration circuit, a 169  $g \cdot L^{-1}$  NaCl solution (Merck). This concentration was chosen in order to minimize the osmotic water 171 transfer as discussed below.' (new lines 161-163)

- 172
- 173
- 174
- 175

176 Line 319: it is not clear how you estimated this 3% and how you distinguished this osmotic water

- 177 transport from the overwhelming electro-osmotic water transport. Is this from Figure 3? Because you
- 178 can't really distinguish between the two modes of water transport during the first part of your

179 desalination. The contribution of osmosis also changes in size and direction throughout the experiment 180 as the salt concentrations change. Is it not simply enough to determine the final volumes in each

181 compartment to account for concentration/dilution of your sample due to water transport?

182 Authors:

183 We estimated this maximal 3% based on the observed osmotic water loss in the end of the desalination 184 when the concentration difference of salt between sample and concentration circuit was the highest and 185 no electro-osmotic water transport occurred simultaneously. However, you are right by saying that the 186 size and direction of osmosis is changing throughout the experiment and we cannot distinguish between 187 the two modes of water transport during the first part of our desalination. The real water loss by osmosis 188 should be definitively lower than 3%. Because of this, we followed the recommendation and eliminated 189 this information from the main text.

190

**Line 351-352: transport of organics in presence of high salt concentrations is expected to be minimal, as demonstrated in a paper by Vanoppen et al. (2015).DOI10.1021/es504389q**

Authors: This reference gives a good comparison for the recovery rates of neutral organics and the neutral
 monosaccharides which were presented in this study. We included this reference by adding this sentence,
 which reads: 'This is in agreement with Vanoppen et al. (2015) who concluded that diffusion and affinity

196 for the membrane are the main drivers for losses of uncharged, low-molecular organics during a 197 desalination using an ED system.' (new lines 360-362)

197 198

199 Line 366-368: this statement is odd here and would be expected more at the end of the introduction.

Authors: We removed this statement from the 'Results and Discussion' chapter and moved it to the introduction. Now the new end of the introduction reads: 'This method with a low need of consumables allows the analysis of monosaccharides with (CCHO) and without hydrolysis (DFCHO), including the possible determination of free amino sugars and free uronic acids. This developed technique was applied to analyze a diverse set of carbohydrates in different kinds of ambient seawater samples.' (new lines 117-120)

206

Figure 4: describe the difference between the full and dotted line in the caption. Please discuss the implications of the dotted line in the discussion. Is this a good quantification of the difference between

- 209 both methods?
- 210 Authors:
- 211 -caption:

212 We added the meaning of the full and the dotted line in the caption of Figure 4. The added line reads: 'The

- full line represents the line of equality. The dotted line represents the regression line between the data of both methods.'
- 215 -discussion in the main text:

216 We added a short discussion about the implication of the dotted line. The added sentence reads: 'The 217 similarity of the equality line and the regression line ( $R^2$ =0.89) using all sugar data indicate a good overall

- agreement between both methods.' (new lines 399-400)
- 219 -good quantification?:
- 220 There exist several statistical methods in order to evaluate the comparability of two methods. You are
- right by asking if a scatter plot and a regression line (Figure 4) between both method's values is the best
- solution, since it suffers certain weaknesses. Therefore, we considered using a Bland-Altman plot (as it was
- recommended for a method comparison by van Stralen et al., 2008). It is a scatter plot in which the
- 224 mathematical difference between the paired measurements is plotted against their average. Usually

225 additional lines represent the mean and the mean±2·standard deviations in order to achieve limits of 226 agreement. The limits of agreement always need to be examined in respect to the scale of the x-axis. A 227 Bland-Altman plot for our data is shown below. Overestimations of rhamnose, arabinose and 228 underestimations of glucosamine can be seen in the Bland-Altman plot, as it is evident in Figure 4 of the 229 manuscript as well. However, we found that the Bland-Altman plot appeared not suitable for the 230 presentation of our data, since its interpretation is not intuitive and needs many additional explanations. 231 The most important observations are described in the main text in any case. Considering the pros and 232 cons, we decided that the correlation plot, together with the explanations in the text, might be an 233 appropriate way to represent our method comparison.

249 Technical corrections:

250 Generally, I propose to introduce the abbreviation ED for electro-dialysis and using it throughout the 251 manuscript.

252 Authors: We introduced the abbreviation 'ED' for 'electro-dialysis'/'electrodialysis' throughout the 253 manuscript.

254

Line 98, replace ',' with 'and' (and there are of course many more examples of ED application)

Authors: In order to express that the mentioned examples only represent a selection of ED applications, we added the term 'amongst others'. The sentence now reads: 'Amongst others, ED is being used for the desalination of salty water to generate potable water and the denitrification of wastewater and soil remediation...' (new lines 97-98)

260

Line 199: sometimes you use 'electro-dialysis'and sometimes 'electrodialysis'. The latter is more frequently used and you can introduce the abbreviation as suggested before.

263 Authors: As already suggested before, we introduced the abbreviation 'ED' for 'electro-264 dialysis'/'electrodialysis' throughout the manuscript.

265

- 266 Authors:
- 267 Line 210: 'slide'= 'slight'
- 268 Authors: We replaced 'slide modifications' with 'slight modifications'. (new line 226)
- 269 Additional changes
- 270 We replaced 'combined to' with 'combined with' (title).
- 271 We added 'hexoses, pentoses' to line 39.
- 272

**273 Cited references:**

274 Cowan, D. A.: Research in the problem of scaling of electrodialsis demineralizers, Texas Water Comm.
275 Bull., 6206, 29, 1962.

Galama, A. H., Saakes, M., Bruning, H., Rijnaarts, H. H. M. and Post, J. W.: Seawater predesalination with
electrodialysis, Desalination, 342, 61–69, doi:10.1016/j.desal.2013.07.012, 2014.

Galier, S., Courtin, M. and Balmann, H. R.: Influence of the Ionic Composition on the Demineralisation of
 Saccharide Solutions by Electrodialysis, Procedia Engineering, 44, 826–829,
 dai:10.1016 /i process 2012.08.587, 2012

- 280 doi:10.1016/j.proeng.2012.08.587, 2012.
- Han, L., Galier, S. and Roux-De Balmann, H.: Ion hydration number and electroosmosis during
  electrodialysis of mixed salt solution, Desalination, 373, 38–46, doi:10.1016/j.desal.2015.06.023, 2015.

Han, L., Galier, S. and Roux-De Balmann, H.: A phenomenological model to evaluate the performances of
 electrodialysis for the desalination of saline water containing organic solutes, Desalination, 422, 17–24,
 doi:10.1016/j.desal.2017.08.008, 2017.

Martí-Calatayud, M. C., García-Gabaldón, M. and Pérez-Herranz, V.: Mass Transfer Phenomena during
 Electrodialysis of Multivalent Ions: Chemical Equilibria and Overlimiting Currents, Applied Sciences, 8(9),
 1566, doi:10.3390/app8091566, 2018.

- Ottosen, L. M., Hansen, H. K. and Hansen, C. B.: Water splitting at ion-exchange membranes and
   potential differences in soil during electrodialytic soil remediation, Journal of Applied Electrochemistry,
   30(11), 1199–1207, doi:10.1023/A:1026557830268, 2000.
- van Stralen, K. J., Jager, K. J., Zoccali, C. and Dekker, F. W.: Agreement between methods, Kidney
  International, 74(9), 1116–1120, doi:10.1038/ki.2008.306, 2008.
- Vetter, T. A., Perdue, E. M., Ingall, E., Koprivnjak, J.-F. and Pfromm, P. H.: Combining reverse osmosis and
   electrodialysis for more complete recovery of dissolved organic matter from seawater, Separation and
- electrodialysis for more complete recovery of dissolved organic matter from seawater, Separation and
   Purification Technology, 56(3), 383–387, doi:10.1016/j.seppur.2007.04.012, 2007.
- 297

298